# Relationship between Self-Esteem and Technological Readiness: Mediation Effect of Readiness for Change and Moderated Mediation Effect of Gender in South Korean Teachers

**DOI:** 10.3390/ijerph19148463

**Published:** 2022-07-11

**Authors:** Jungsug Kim, Eunjeung Kim

**Affiliations:** 1Department of French Language Education, Seoul National University, Seoul 08826, Korea; aramchi@snu.ac.kr; 2Department of Home Economics Education, Kyungpook National University, Daegu 41566, Korea

**Keywords:** self-esteem, technology readiness (TR), readiness for change (RC), gender, moderated mediation effect, Korean teachers

## Abstract

This study aimed to investigate the moderated mediation effect of gender and the mediation effect of readiness for change (RC) on self-esteem and technology readiness (TR) among South Korean teachers. Participants were 302 teachers who are living and working in South Korea. The collected data were analyzed using frequency and correlation analyses, and the moderated mediation effect. First, we considered the differences in TR and RC according to gender, and they were reported to be higher in men than in women. The number of teachers who had worked for more than 11 years was significantly higher than that of other teaching career groups. Second, correlation analysis showed a positive correlation between self-esteem, TR, and RC by gender. Third, teachers’ RC mediates the relationship between self-esteem and TR. Fourth, the teachers’ gender moderated the relationship between self-esteem and RC. Additionally, teachers’ gender moderated the mediating effect of RC on self-esteem and TR. Finally, based on the study results, we suggest the development of a program for improving self-esteem to enhance TR and RC among teachers of both genders. Additionally, future research should consider universal teacher sampling to facilitate the comparison of teachers’ characteristics and to identify variances in the data.

## 1. Introduction

Most studies targeting teachers explore their roles and competencies. However, given the unexpected changes the coronavirus disease (COVID-19) pandemic has induced, when coupled with the present state of super-connectedness due to information and technology advances amidst the Fourth Industrial Revolution, teachers are experiencing severe stress [1]. Their stress is intensified by virtual classes and blended learning, using digital devices and other new technologies [2]. Under these circumstances, only teachers who are fully prepared for change, particularly technologically driven change, can positively cope with critical situations.

In this context, teachers’ readiness (TR) refers to individual differences regarding the acceptance and utilization of technology, that is, one’s personal tendency toward technology acceptance than the ability to accept new technologies. TR is classified into positive emotions toward technology, such as optimism as well as innovativeness, and negative emotions, such as discomfort and insecurity, which negatively affect technological acceptance and could work as a psychological factor influencing the utilization of technologies [3]. Specifically, compared with people who have primarily positive emotions, those whose TR is characterized by negative emotions are highly susceptible to stress. Most TR research is related to the conditions required for healthy organizational growth. However, the ongoing pandemic, a crisis that requires teachers to use technology, has accelerated the urgent need for research on teachers’ TR.

Readiness for change (RC) is related to psychological tension and stress levels [4,5]. Although most RC research pertains to corporate resistance to change, research reports on how organizational members’ RC could lead to positive organizational changes has highlighted the importance of the concept itself [6]. The education system demands teachers with technical abilities to freely use technology on- and off-line, a requirement that underscores the importance of possessing RC, that is, positively perceiving and participating in change. Conversely, individuals who are resistant to change or who have low RC levels may experience psychological stress. Self-esteem is known to function as a psychological stress management mechanism that decreases stress levels in relation to individuals’ RC [7,8]. Research has shown that individuals with high self-efficacy who believe that they are equipped with the technological skills and expertise necessary to cope with change are more likely to approach change positively [9].

Although much research has been conducted regarding change, the correlations between teachers’ self-esteem, TR, and RC and gender remains relatively unexplored. Thus, this study aimed to analyze the effectiveness of RC with respect to the effects of self-esteem and RC on gender. By understanding these relationships, we sought to identify the mediation effect of RC between self-esteem and TR among teachers, and explore the moderated mediation effect of gender on the mediation effect of RC between self-esteem and TR among the same participants. The research results are expected to provide basic data confirming the necessity of a support system to help teachers adapt positively to educational changes and to help them become more competent, according to their respective gender. This enhances the quality of education.

## 2. Literature Review

### 2.1. Technology Readiness

Owing to advancements in artificial intelligence, computers, and cloud technology, modern society is becoming well-connected and extremely intelligent. Against this backdrop, people who are unprepared for the advent of these technologies experience stress and exhibit negative behaviors when placed in situations that require the use of advanced technologies [10]. Parasuraman [3] presented TR as a predictor of consumption behaviors by understanding consumers’ responses to the latest technologies in the context of a consumer–technology relationship. It was characterized by the motivators of optimism and innovativeness, which exert positive effects on technology acceptance, and the inhibitors of discomfort and insecurity, which exert negative effects. TR measures individuals’ intrinsic tendency to adopt technologies, rather than the possibility of using new technological products [11,12]. For instance, it is important to offer products with utility by bolstering technological stability and security to attract people with high technological anxiety, and also to develop products with simple functions to draw people who feel uncomfortable using technology. This type of TR research was conducted to develop consumer products. Their results show that when consumers perceive technology positively, they have high TR levels and use intentions [13].

Such studies also show that TR motivators and inhibitors are positively and negatively related to educational level, respectively [14,15]. Specifically, regarding perceptions of technology comprising TR, people with lower education level are associated with low optimism and innovativeness, strong discomfort, as well as insecurity. Blut and Wang [16] reported significant effects of TR on technological fear. Evidently, TR is differentiated from context-specific behavioral attitudes or beliefs in one’s ability to use technology, and is distinct from an individual’s general innate attitude [3,17]. Therefore, teachers in South Korea may be required to accept and apply technologies in situations differentiated by personal characteristics, as noted in the extant research. However, most of the existing research has involved kindergarten or elementary school teachers, who possess relatively high overall TR levels, with differences in innovativeness according to teaching experience. However, despite the ongoing COVID-19 pandemic, which demands innovative application of technologies on education, research on secondary school teachers’ TR is scarce. Given that, in the educational context, teachers’ TR is related to technological fear, and teachers’ negative emotions can lead to stress; gaining an understanding of current teachers’ perceptions of TR is essential.

### 2.2. Readiness for Change

Armenakis et al. [18] defined RC as a cognitive precursor to behavior, and classified it into cognitive, affective, and behavioral dimensions. The cognitive dimension refers to the knowledge structure of beliefs, thoughts, and perceptual responses and changes, while the affective dimension is related to emotions linked to change, and the behavioral dimension discusses actual behavior [19,20,21]. Here, “readiness” is a mindset shared among the members of an organization amidst change, which comprises beliefs, attitudes, and individuals’ intentions regarding the degree of necessary change, and the organization’s ability to successfully implement them.

Although RC research focuses mainly on individuals or organizations, this study aimed to analyze the mediation effects of teachers’ RC on the relationship between self-esteem and TR to support discussions on teachers’ RC at an individual level. Jeong and Choi [20] defined individual RC as intentional readiness, which can be interpreted as a series of cognitive change processes involving willful effort to participate in change after an individual recognizes the importance and necessity of such change and formulates positive emotions about it. Regarding individual-level change, some research shows positive effects of characteristics such as self-efficacy, ability to work, educational attainment, and position on RC [20,22]. Moreover, RC has positive effects on an individual’s ability to overcome psychological problems [23,24]. Based on these results, we could infer that individual teachers’ RC as they cope with pandemic-induced changes in education may be related to stress. Specifically, a teacher with a positive attitude toward change can summon both the willpower to support change and the confidence that they will succeed under changing circumstances, thereby reduce psychological stress [25].

### 2.3. Relations between Self-Esteem, TR, and RC on Gender

Self-esteem encompasses all positive and negative self-evaluations related to perceiving oneself as a respectable and valuable person and is therefore an important factor for psychological adaptation [26,27]. Wylie [28] noted that self-esteem is influenced by others’ opinions and is regarded as a meaningful, subjective evaluation of self-value and environmental factors. It is an important aspect of human behavior and influences the achievement related to self-realization and social life, including interpersonal relationships. Furthermore, research has shown that high self-esteem plays a mediation role, enabling more efficient stress management [7,29,30,31,32,33]. This finding implies that individual teachers’ RC and TR are related to self-esteem and that self-esteem has the potential to reduce stress [23]. Research also indicates that teachers’ self-esteem influences their teaching efficacy and job satisfaction [34], which increases with age and professional experience [32,35]. Since the COVD-19, teachers have perceived difficulty in transitioning to online learning. The results of the research which shows that RC has a mediation effect on self-esteem, behaviors of trying and analyzing organizational value, or adaptive performance mean that RC should be considered in the process of change acceptance, work behavior, or adaptive performance of self-esteem [36,37]. Additionally, there are studies reporting that women show more negative technology readiness than men [38], while other studies show that there is no relationship between RC and gender [39].

These results point to the necessity of developing varied measures to support the growth of teachers’ self-esteem and, in turn, their teaching efficacy with age and professional experience. Based on the aforementioned studies, self-esteem is expected to exert a significant effect on the relationship between TR and RC. TR also influences RC and reduces stress. Exploring these relationships, this study aimed to investigate the mediation effect of RC between self-esteem and TR as well as the moderated mediation effect of gender on the relationship between self-esteem and RC in Korean teachers.

Hence, the following research questions were set:Are there differences in the main variables based on sociodemographic characteristics?Are there correlations among the main variables of the study?Does teacher’ RC mediate the relationship between self-esteem and TR?Do teachers’ gender moderate mediation effect between self-esteem and TR?

This study examined the positive effects of RC on teachers, and whether RC contributes to increasing self-esteem and TR. The research findings may have implications for policy recommendations aimed at improving self-esteem, TR, and RC depending on the gender of the teachers.

## 3. Materials and Methods

### 3.1. Research Model

Based on previous research, a research model for the mediation effect of RC and the moderated mediation effect of gender on the relationship between self-esteem and TR was established (Figure 1).

### 3.2. Participants

Participants were 302 randomly selected secondary school teachers (middle school to high school), representing 0.078% of the total secondary school teacher population in the Republic of Korea (389,099 teachers: 218,895 females; 170,204 males). We followed Chou and Bentler’s [40] recommendation and used 200 cases as a reasonable sample size to meet the requirements of this study. Data were collected using Google’s mobile URL from October 2021 to November 2021. We drafted the questionnaire in consultation with teachers who were active in the teacher-learning community. After conducting an online survey with 320 teachers, 308 questionnaires were collected. Among the survey respondents, the analysis was limited to the teachers in charge of secondary school subjects.

Participation was voluntary and all participants were assured anonymity and provided verbal or written consent.

A total of 302 completed questionnaires were used for the final analysis. Six surveys, including insincere responses were excluded, and the remaining 302 were used finally. Convenience sampling was applied at secondary schools across the Republic of Korea, and 302 teachers (115 males, 37.7%; 187 females, 61.9%) were selected as subjects whose data were used in the final analysis. Regarding the participants’ teaching experience, 115 had less than 10 years of service (38.1%), while 187 had more than 10 years of service (61.9%).

### 3.3. Measurement Tools

This study adopted Lee et al.’s [12] adaptation of Rosenberg’s [36] 10-item self-esteem scale. Items were measured on a 5-point Likert scale ranging from 1 (not at all) to 5 (very well). Items 3, 5, 8, 9, and 10 are recorded. The scale’s reliability in this study was established based on Cronbach’s alpha of 0.852. Five items were recorded. A high total score indicates high self-esteem.

This study adopted Parasuraman’s [3] TR index (TRI), which is a 13-item scale that measures optimism, innovativeness, discomfort, and anxiety. The items included “I like to use the newest technology,” and “When the latest technology comes out, I tend to be the first to accept it among my friends.” Items were scored on a 5-point Likert scale ranging from 1 (not at all) to 5 (very well). Items 8–13, with opposite meanings, were recorded. The higher respondent’s score indicates their TR is higher. The reliability of the TRI in this study was established based on a Cronbach’s alpha of 0.869. Six items of discomfort and anxiety were recorded. A high total score indicated high TR.

To measure cognitive, emotional, and intentional RC, we used Jeong and Choi’s [20] adaptation of Bouckenooghe et al.’s [41] 11-item RC scale. Example items included, “Most changes will have a negative impact on us,” and “Life will be improved by change.” Items were scored on a 5-point Likert scale ranging from 1 = Not at all to 5 = Very well. The higher the respondent’s score, the higher is the RC. The reliability of the RC scale in this study was established based on Cronbach’s alpha of 0.909. Four items were recorded. A high total score indicates a high RC.

### 3.4. Data Analysis

The collected data were analyzed using SPSS Win 26.0, and SPSS Macro PROCESS. Correlation analysis was performed to identify correlations between the main variables. Hayes’s [42] PROCESS macro models 4 and 7 were used to identify the moderated mediation effect via bootstrapping with 5000 samples and a 95% confidence interval. The independent variable (X) and moderator variable (W) were mean-centered prior to the analysis.

## 4. Results

### 4.1. Differences According to Sociodemographic Characteristics

As a result of examining the differences in the main variables according to gender, teaching career was found to be statistically significant (Table 1). First, considering the differences in TR and RC according to gender, TR and RC were reported to be higher in men than in women. However, there were no significant differences in self-esteem or gender. Second, compared to teaching careers, the number of teachers who had worked for over 11 years was significantly higher than that of other teaching career groups.

### 4.2. The Moderated Mediation Effect of Gender on the Relationship between Self-Esteem and TR

Moderated mediation effects refer to changes in the intensity or direction of the mediation process according to the level of the moderated mediation variable. Statistically, this implies conditional indirect effects [42]. To verify the moderated mediation effects, the preceding hypotheses should be met: the mediation effects of RC on the effects of self-esteem on teachers’ TR are significant (MACRO model 4), and the moderated mediation effects of gender on the effects of self-esteem on RC are significant (MACRO model 7). Therefore, to verify the moderated mediation effects, the two hypotheses were analyzed, and consequently, the matter of satisfying the preceding hypotheses was verified.

#### 4.2.1. Correlation of Main Variables

To examine the correlation between the main variables, this study conducted a correlation analysis and found that the correlations between all variables were statistically significant (Table 2 and Table 3). The results showed that self-esteem and RC (r = 0.533, *p* < 0.01), RC and TR (r = 0.509, *p* < 0.01), self-esteem and TR (r = 0.305, *p* < 0.01), and TR and gender (r = 0.211, *p* < 0.01) were positively correlated (*p* < 0.01). For male teachers, self-esteem and RC showed the highest correlation (r = 0.667, *p* < 0.01), followed by RC and TR (r = 0.477, *p* < 0.01) and self-esteem and TR (r = 0.392, *p* < 0.01).

For female teachers, self-esteem and TR showed the highest correlation (r = 0.525, *p* < 0.01), followed by self-esteem and RC (r = 0.432, *p* < 0.01) and self-esteem and TR (r = 0.250, *p* < 0.01). In other words, when teachers’ self-esteem was high, both RC and TR were high. Additionally, when RC was higher, TR was also high. The correlation coefficients ranged from 0.250 to 0.667, and there was no multicollinearity of the major variables.

#### 4.2.2. Mediation Effect of Readiness for Change and Moderated Mediation Effect of Gender

This study verified the moderated mediating effects of gender on the effects of self-esteem on teachers’ TR through the mediation of RC. For this, using the SPSS Process Macro No.7 Model presented by Hayes [42], bootstrapping analysis was conducted with a total of 5000 samples, confidence intervals of 95%, and gender. Because gender as a moderated mediation variable was a nominal variable, mean centering was not conducted. The results are presented in Table 4 and Table 5 (Figure 2 and Figure 3). The moderated mediation effect of gender on the effects of self-esteem on teachers’ TR is shown in Table 4. The interaction between self-esteem and teachers’ gender had positive effects on RC (Coeffect = 0.331, *p* < 0.010). In other words, RC also increased when self-esteem increased. The influence of self-esteem on RC increased according to gender.

In addition, regarding the conditional indirect effects of gender, moderated mediation effects were significant for both men and women. In other words, “0” was not included between Boot LLCI and Boot ULCI in the 95% confidence interval; therefore, the conditional indirect effects were statistically significant for both men and women. This means that when the self-esteem of male and female teachers is high, RC also increases. As shown in Figure 3, female teachers showed a rapid slope compared to male teachers, which showed an increased influence of self-esteem on RC.

Next, in the results of examining the moderated mediation effects of gender on the path of self-esteem→RC→TR, the interaction between self-esteem and gender of teachers had positive effects on RC, and RC had positive effects on TR (Coeffect = 0.513, *p* < 0.001). These results show that the effects of self-esteem on TR through the mediation of RC could change depending on the gender.

To verify the size or direction of moderated mediation effects that would mean changes in mediation effects according to the level of moderated mediation variable, this study verified the conditional indirect effects by gender on the effects between self-esteem and TR by passing through RC (see Figure 2 and Figure 3). As shown in Table 5, moderated mediation effects were significant for both male and female teachers. In other words, “0” was not included between Boot LLCI and Boot ULCI in the 95% confidence interval; therefore, the conditional indirect effects were statistically significant. This means that when the self-esteem of male and female teachers is higher, RC increases, and the increased RC raises TR.

Additionally, in the index of moderated mediation effects of gender, “0” was not included between Boot LLCI and Boot ULCI in the 95% confidence interval (Table 5); therefore, the significance of the moderated mediation effect of gender on the mediation effect of RC on self-esteem was verified.

## 5. Discussion

This study aimed to investigate the mediation effect of RC between self-esteem and TR, and the moderated mediation effect of gender on the relationship between self-esteem and RC in Korean teachers. To this end, 302 teachers across the country were surveyed between October 2021 and November 2021. The results are discussed below.

First, considering the differences in TR and RC according to gender, TR and RC were reported to be higher in men than in women. The number of teachers who had worked for over 11 years was significantly higher than that of other teaching career groups [38].

Second, correlation analysis showed a positive correlation between self-esteem, TR, and RC on gender. Therefore, to raise teachers’ TR, increasing their self-esteem and RC levels is important. These findings align with those of Jung et al. [13], who reported that TR affects technological fear [16], and is positively correlated with positive perceptions of technology and TR. The findings also align with research indicating that the incidence of stress among teachers has increased after the pandemic owing to their digital gap. Moreover, the findings of this study are consistent with previous research which suggest that individuals’ self-esteem and RC exert positive effects on self-efficacy [20,22], and the ability to overcome stress [7,23,24,29,30,32].

Third, teachers’ RC mediates the relationship between self-esteem and TR. Furthermore, no significant impact was found between self-esteem, TR and RC or self-esteem and RC. Additionally, gender had a moderated mediation effect on the relationship between self-esteem and TR, whereas RC was found to play a mediating role between self-esteem and TR. This is consistent with previous research that found an increased TR is followed by an increased RC [16,23,24]. Therefore, implementing an adequate educational support system to boost teachers’ self-esteem and encouraging enhancement of RC to increase TR are necessary. Conversely, regarding the mediating effect of RC, the interaction effect of TR (independent variable) and gender (moderated mediation variable) was found to significantly affect RC (mediated variable), suggesting that RC mediates the relationship between self-esteem and TR. The interaction between self-esteem and gender of teachers had positive effects on RC, and RC had positive effects on TR (Coeffect = 0.331, *p* < 0.010).

This study verified the conditional indirect influence of gender on the effects of self-esteem on TR through RC. Although self-esteem does not directly affect TR, the mediation effect of RC has a positive relationship with self-esteem and TR depending on gender, which is consistent with previous research findings [7,29,30,32,36,37]. This highlights the need for support to improve self-esteem and RC, in addition to psychological support to boost self-esteem, to raise teachers’ TR amidst the changing educational landscape since the onset of the pandemic.

Fourth, teachers’ gender moderated the mediating effect on the relationship between self-esteem and RC. Additionally, teachers’ gender moderated the mediating effect of RC on self-esteem and TR. An examination of the moderated mediation effects of gender on the path of self-esteem→RC→TR showed that the interaction between self-esteem and gender of teachers had positive effects on RC, and RC had positive effects on TR (Coeffect = 0.513, *p* < 0.001). These results show that the effects of self-esteem on TR through the mediation of RC could change depending on the gender. These results show that institutional support for the improvement of self-esteem is more important for female teachers than for male teachers.

Finally, based on the study results, we suggest the development of a program to improve self-esteem and enhance TR and RC among teachers. Additionally, future research should consider universal teacher sampling to facilitate the comparison of teachers’ characteristics and to identify variances in the data.

## 6. Conclusions

This study examined the relationships between self-esteem, TR, and RC in Korean teachers and found that RC mediates the relationship between self-esteem and TR, and gender exerts a moderated mediation effect on the impact of RC on self-esteem. Such results mean that even though self-esteem does not directly affect TR, it can increase RC, implying a raise in the level of TR. Despite the increased importance of TR owing to technological changes, it is crucial to raise the level of RC first. Therefore, providing policies that support teachers in coping with changes through the teacher learning community or training opportunities to prepare them continuously for changes in the school field is necessary.

Additionally, the result of the study in which gender has effects on self-esteem and RC indicates the necessity to have differences in political support depending on gender. In other words, for female teachers, it is necessary to provide institutional support for the improvement of self-esteem by raising their RC rather than simply raising the level of TR.

Our study has the following limitations that offer scope for future research: First, this study has convenience sampling limitations as it only included teachers who voluntarily participated in the survey. In the future, universal teacher sampling will be necessary to facilitate the comparison of teachers’ characteristics and identify variances in the data. Second, amid several changes in education in South Korea, teachers are required to equip themselves with educational competencies that reflect technological advances. Grappling with this adaptive process may increase stress among teachers, necessitating the establishment of gender-based measures to boost self-esteem and RC to improve TR. These measures should be combined with institutional support to enhance teacher TR. The development and implementation of programs to enhance self-esteem, TR, and RC based on teachers’ characteristics are urgent tasks. Studying the effects of self-esteem programs on TR levels among vulnerable groups of teachers focusing on gender is also crucial.

Third, this study only examined teachers’ TR in relation to self-esteem, RC, and gender, but TR among teachers should be studied in the future to inform the development of this support program.

Despite the aforementioned limitations, this study is the first to elucidate the moderated mediation effect of gender in relation to teachers’ TR. Therefore, this study makes a meaningful contribution by providing new basic data to inform the development of self-esteem and the effect of gender as a mechanism for increasing RC and TR among Korean teachers.

## Figures and Tables

**Figure 1 ijerph-19-08463-f001:**
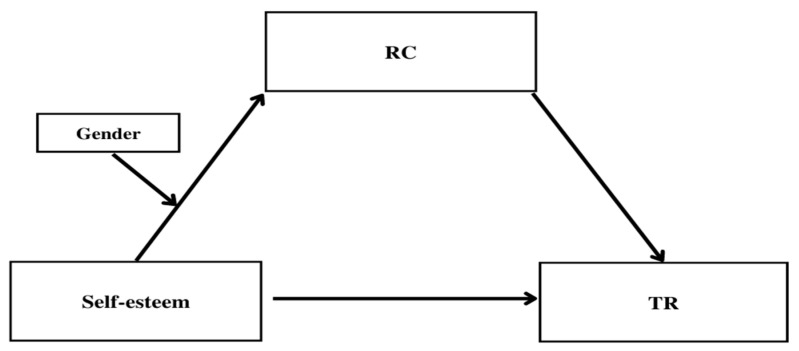
Research Model.

**Figure 2 ijerph-19-08463-f002:**
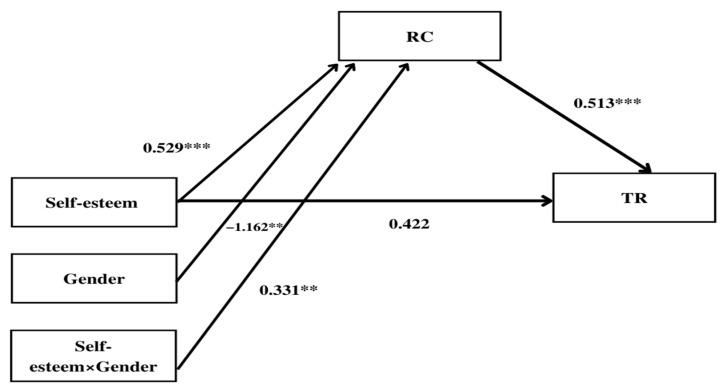
Statistical Figure on Moderated Mediation Effects of Gender. ** *p* < 0.01, *** *p* < 0.001.

**Figure 3 ijerph-19-08463-f003:**
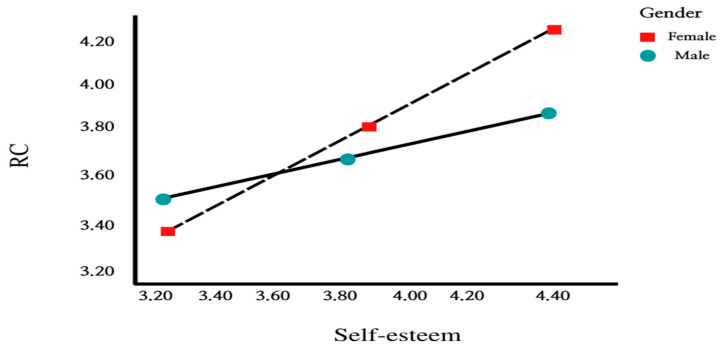
Moderated mediation effect of self-esteem.

**Table 1 ijerph-19-08463-t001:** Differences according to sociodemographic characteristics.

IV			N	M (S.D.)	t/F
Gender	Self-esteem	male	115	3.84 (0.586)	1.87
		female	187	3.84 (0.554)	
	TR	male	115	3.44 (0.654)	4.80 *
		female	187	3.18 (0.535)	
	RC	male	115	3.81 (0.637)	5.51 *
		female	187	3.70 (0.505)	
Teaching career	Self-esteem	1–10	115	3.73 (0.642)	4.81 *
		over 11	187	3.90 (0.504)	

* *p* < 0.05.

**Table 2 ijerph-19-08463-t002:** Correlation between variables (N = 302).

	Self-Esteem	TR	RC
Self-esteem	1		
TR	0.305 **	1	
RC	0.533 **	0.509 **	1
gender	0.000	0.211 **	0.094

** *p* < 0.01.

**Table 3 ijerph-19-08463-t003:** Correlation between variables by sex (N = 302, male = 115, female = 187).

		Self-Esteem	TR	RC
Male	Self-esteem	1		
	TR	0.392 **	1	
	RC	0.667 **	0.477 **	1
Female	Self-esteem	1		
	TR	0.250 **	1	
	RC	0.432 **	0.525 **	1

** *p* < 0.01.

**Table 4 ijerph-19-08463-t004:** Moderated mediation effect of gender on the relation between self-esteem and TR.

Path		Mediation Variable Model (DV:RC)			Independent Variable Model (DV:TR)		
		Coeffect	SE	T Value	Coeffect	SE *β*	T Value
constant		1.712	0.188	9.114 ***	1.167	0.508	5.084 ***
ID	Self-esteem	0.529	0.048	10.915 ***	0.050	0.062	0.422
Mod var. (W)	gender	−1.162	0.374	−3.106 **	-	-	-
Interaction	Self-esteem × gender	0.331	0.096	3.434 **			
Test of highest order unconditional interaction							
Interaction		*R*^2^-change	F				
TR × Self -esteem		0.027	11.795 **				
Mediation	RC	-	-	-	0.513	0.062	8.220 ***
Model summary	*R^2^*	0.320			0.260		
	F	46.762 ***			11.795 **		
Conditional effects of TR at values of gender							
gender	effect	SE	t		*p*	LLCI ^a^	ULCI ^b^
Female	0.202	0.050	6.407		0.000	0.273	0.515
Male	0.355	0.043	8.214		0.000	0.579	0.871

** *p* < 0.01, *** *p* < 0.001. ^a^: LLCI = lower limit of indirect effect within the 95% confidence interval. ^b^: ULCI = The higher limit of the indirect effect within the 95% confidence interval.

**Table 5 ijerph-19-08463-t005:** The conditional direct and indirect effects of self-esteem on TR.

Direct Effect (Self-Esteem → TR)					
Effect	SE	t	*p*	LLCI	ULCI
0.050 *	0.062	0.804	0.422	–0.072	0.172
Conditional indirect effect (Self-esteem → RC→TR)					
Gender	Effect	BootSE	BootLLCI	BootULCI	
Female	0.202	0.049	0.115	0.307	
Male	0.372	0.066	0.242	0.505	
Index of moderated mediation					
	Index	BootSE	BootLLCI	BootULCI	
Gender	0.170	0.060	0.058	0.294	

* *p <* 0.05.

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
