# Peer review of "Relationship between Self-Esteem and Technological Readiness: Mediation Effect of Readiness for Change and Moderated Mediation Effect of Gender in South Korean Teachers"

_ijerph, 2022, doi:10.3390/ijerph19148463_

Round 1

Reviewer 1 Report

The authors made good work in improving the manuscript. 

I wish them good luck with their research.

Author Response

Thank you again for your scholarly assistance and encouragement, particularly the reviewers who paid careful attention to the text and provided detailed suggestions to improve the manuscript substantively.

Reviewer 2 Report

Dear authors

I hope you are well and in good health.

I find your manuscript interesting in terms of the approach to the relationship between the variables, but I am concerned that some things are not clear enough to adequately evaluate your work after the modifications made:

1- The title of the manuscript changed to "Relationship Between Self-Esteem and TR: Mediation Effect of RC and Moderated Mediation Effect of Gender in South Korea Teachers", however, the abstract, introduction and literature review were not modified (except for the exclusion of the section dealing with depression) so that the coherence between the new title and these sections is lost. For example, the abstract begins with the objective of the study (rows 10-12), in which the variable "depression" is mentioned, which is not taken into account later in the methodology.

2- The introduction and the literature review do not have major changes, so it is still oriented to the study of depression in Korean teachers, however, this does not seem to be the aim of the study after the modifications made by the authors to the manuscript.

3- The aim/objective of the work is presented on several occasions, but they are not consistent in all cases, for example, in the abstract it is stated: "to investigate the moderating mediation effect of self-esteem on the mediation effect of readiness for change (RC) on technology readiness (TR) and depression in teachers" (rows 10-12). Then, "this study sought to investigate the mediation effect of RC between TR and depression among teachers, identify the moderating effect of self-esteem between TR and RC, and investigate the moderating mediation effect of self-esteem on the mediation effect of RC on depression in teachers" (rows 63-66), then: "this study aimed to analyze the mediating effects of teachers' RC on the relationship between TR and depression" (rows 119-121).  Later the variable "depression" is excluded, and the variable "gender" is included: "this study aims to investigate the mediation effect of RC between self-esteem and TR and the moderated mediation effect of gender on the relation self-esteem and RC in Korean teachers" (rows 159-161). It is necessary to standardize the aim/objective throughout the paper.

4- The variable "depression" is mentioned in the research model (rows 174-176), but it is not shown in the actual figure, nor is it mentioned how it would be assessed. Even the section dealing with depression in the literature review was eliminated, so it could be inferred that this variable would not be taken into consideration in the study.

5- It is important that the paper is homogeneous in the use of the terms "mediation effect", "moderation effect", "moderated mediation effect", since in some passages "mediation" and "moderation" are used synonymously, if there is any difference between these terms in their statistical or theoretical treatment, describe it adequately in your manuscript.

6- The results of the correlation analysis do not provide information to say that TR and gender were positively correlated (rows 256) since only the sample was divided by gender, but the actual variable “gender” was not correlated with any variable.

7- Figure 3 is not mentioned in the text.

8- Table 4 presents results of the direct and indirect conditional effect of TR on depression, but the role of depression or how it was assessed is not specified in the methodology. Even the section dealing with depression in the literature review was eliminated, so it could be inferred that this variable would not be taken into consideration in the study.

9- In the discussion, the authors could clarify what they mean by the expression "self-esteem does not directly affect TR, its mitigating effect contributes to more efficient TR among individuals with gender" (rows 352-353).

Author Response

We appreciate your dedicated support and service for the International Journal of Environmental Research and Public Health. In Addition, we would like to express our gratitude for the reviewers’ thoughtful comments regarding our manuscript. They have been a great help in suggesting ideas to improve our manuscript, “Relationship between Self-esteem and Technological Readiness: Mediation Effect of Readiness for Change and Moderated Mediation Effect of Gender in South Korean Teachers” As you will find, we have considered every comment, and we attended to each one, in most cases, making revision to the manuscript. We hope you will concur that the manuscript is significantly strengthened by the revisions, as suggested by reviewers’ insights. We address the reviewers’ comments below, in the order in which they appeared. Thank you again for your scholarly assistance and encouragement, particularly the reviewers who paid careful attention to the text and provided detailed suggestions to improve the manuscript substantively.

  1. The title of the manuscript changed to "Relationship Between Self-Esteem and TR: Mediation Effect of RC and Moderated Mediation Effect of Gender in South Korea Teachers", however, the abstract, introduction and literature review were not modified (except for the exclusion of the section dealing with depression) so that the coherence between the new title and these sections is lost. For example, the abstract begins with the objective of the study (rows 10-12), in which the variable "depression" is mentioned, which is not taken into account later in the methodology. [revised as requested] (12, 32~33, 44, 64, 97, 121122, 144-145, 175, 242, 316, 334, 337, 394, 399)
  2. The introduction and the literature review do not have major changes, so it is still oriented to the study of depression in Korean teachers, however, this does not seem to be the aim of the study after the modifications made by the authors to the manuscript. [done]
  3. The aim/objective of the work is presented on several occasions, but they are not consistent in all cases, for example, in the abstract it is stated: "to investigate the moderating mediation effect of self-esteem on the mediation effect of readiness for change (RC) on technology readiness (TR) and depression in teachers" (rows 10-12). Then, "this study sought to investigate the mediation effect of RC between TR and depression among teachers, identify the moderating effect of self-esteem between TR and RC, and investigate the moderating mediation effect of self-esteem on the mediation effect of RC on depression in teachers" (rows 63-66), then: "this study aimed to analyze the mediating effects of teachers' RC on the relationship between TR and depression" (rows 119-121).  Later the variable "depression" is excluded, and the variable "gender" is included: "this study aims to investigate the mediation effect of RC between self-esteem and TR and the moderated mediation effect of gender on the relation self-esteem and RC in Korean teachers" (rows 159-161). It is necessary to standardize the aim/objective throughout the paper. [revised as requested](10-12, 63-66, 119-121, 159-161)
  4. The variable "depression" is mentioned in the research model (rows 174-176), but it is not shown in the actual figure, nor is it mentioned how it would be assessed. Even the section dealing with depression in the literature review was eliminated, so it could be inferred that this variable would not be taken into consideration in the study. [revised as requested]( 174-176)
  5. It is important that the paper is homogeneous in the use of the terms "mediation effect", "moderation effect", "moderated mediation effect", since in some passages "mediation" and "moderation" are used synonymously, if there is any difference between these terms in their statistical or theoretical treatment, describe it adequately in your manuscript. [revised as requested](10. 13, 65, 166, 175, 224, 247, 275-276, 287, 299, 375)
  6. The results of the correlation analysis do not provide information to say that TR and gender were positively correlated (rows 256) since only the sample was divided by gender, but the actual variable “gender” was not correlated with any variable. [revised as requested](253-260)
  7. Figure 3 is not mentioned in the text. [Done](279)
  8. Table 4 presents results of the direct and indirect conditional effect of TR on depression, but the role of depression or how it was assessed is not specified in the methodology. Even the section dealing with depression in the literature review was eliminated, so it could be inferred that this variable would not be taken into consideration in the study. [revised as requested](285)
  9. In the discussion, the authors could clarify what they mean by the expression "self-esteem does not directly affect TR, its mitigating effect contributes to more efficient TR among individuals with gender" (rows 352-353). [revised as requested] (356-358)
  10. As the reviewer suggested for checking all the statistical digits, we did double check all the data in the table and description.

Thanks again.

Sincerely,

Eun Jeung Kim (Corresponding Author) coronia3@gmail.com

This manuscript is a resubmission of an earlier submission. The following is a list of the peer review reports and author responses from that submission.

Round 1

Reviewer 1 Report

Dear authors

After more than two years since the beginning of the COVID-19 pandemic, the knowledge about the changes it has produced in society is invaluable, so the contributions in this sense have an intrinsic value, however, I find several areas of opportunity in your work:

1- The article has several flaws in terms of formatting, such as several underlined words, citations and references in APA format, and errors in the tables. Please refer to the instructions for authors (https://www.mdpi.com/journal/ijerph/instructions).

2- The document contains unreadable figures due to the quality of the selected images. Please upload better quality images that can be read properly.

3- When authors write "self-esteem affects TR, and TR affects RC. So, self-esteem is expected to exert a significant effect on the relationship between TR and RC. (Rows 175-176) it is clear the need to establish a mediation analysis (Mediation analysis tests a hypothetical causal chain where one variable X affects a second variable M and, in turn, that variable affects a third variable Y) to test the hypothesized relationships between these variables. However, they then continue "TR additionally influences RC and serves to reduce depression" (row 177). Finally, they say "...this study sought to investigate the mediation effect of RC between TR and depression among teachers" (row 178-179). Why then do the authors seek the mediating effect of CR if they previously established that it is TR that influences CR? From the first sentence it seems that self-esteem has an indirect/mediating effect between TR and CR.

4- Related to the previous comment, it is stated that self-esteem mediates the relationship between TR and CR, then it is said that in the analysis the mediator is CR of the relationship of TR and depression, but in Figure 1 it is stated that the mediating variable is "readiness for readiness". What is the actual role of each variable in the research model (Similar to a conditional process model)?

5- The authors state that one of their objectives is to "...investigate the moderating mediation effect of self-esteem on the mediation effect of RC on depression in teachers" (row 180-181), but when they present their research model they say "...the moderating mediation effect of self-esteem in the relationship between TR and depression was established" (row 192-193). It is not clear for me what relationships the researchers were looking for

6- The authors do not present any sociodemographic variable or socioeconomic indicator about the teachers or the geographic areas from which the sample was drawn. Aren't these aspects relevant to know the access to technology in the first place? Could any difference by sex be hypothesized in the results? It would be important to include these elements for comparisons with other populations.

7- It would be helpful to include a perspective on the public policy implications of your results in the conclusion

Reviewer 2 Report

In the summary text, please pay attention to this phrase: "First, correlation analysis showed a positive correlation between TR and RC, TR and depression, and RC and depression. Second, teachers' RC mediated the relationship between TR and depression." In fact, the positive correlation is only between TR and RC (as we see in table no. 1 in the body of the article)

The name of the mediating variable in figure no. 1 is wrong. Readiness for change must be written.

When presenting the instruments, it must be specified how many items were inversely scored, especially since there are contrasting examples of items, for example in RC. It is necessary to specify the significance of a high total score and a low total score of each instrument.

The issue related to the openness of teachers to technology is very current and important. Probably in the future, specific self-efficacy for the use of technology would be a better indicator than self-esteem.

Reviewer 3 Report

Dear authors,

The authors conducted a study that aimed to test the moderating mediation effect of self-esteem on the mediation effect of readiness for change (RC) on technology readiness (TR) and depression. The authors used a sample of teachers from South Korea.

Despite the positive features of the study, there are some considerations to take care before the paper is published. 

The following comments will summarize my appreciation and major concerns with your paper. I hope these comments help you further improve your study.

Theoretical background

1.     The literature review is adequate and suits the main goals of the paper.

2.     What are the main goals of the paper? What do you want to achieve? And what do you add to the literature.

Method

3.     Did the authors control for any variables?

4.     Please add more information regarding the data collection procedure. How did the teachers were recruited? What is the response rate?

Discussion

 The discussion was short and did little more than summarise your findings. 

5.     Please, develop the discussion section, considering the theoretical implications.

6.     The limitations and future research should be elaborated.

7.     What are the main practical implications of the study? And what do you add that is not tested before?